# Sourdough Fermentation as a Tool to Improve the Nutritional and Health-Promoting Properties of Its Derived-Products

Carla Graça [1,2], Ana Lima [3], Anabela Raymundo [1] and Isabel Sousa [1,*]

1   Linking Landscape Environment Agriculture and Food (LEAF) Research Center,
    Instituto Superior de Agronomia, Universidade de Lisboa, Tapada da Ajuda, 1349-017 Lisbon, Portugal;
    carlalopesgraca@isa.ulisboa.pt (C.G.); anabraymundo@isa.ulisboa.pt (A.R.)
2   Department of Food and Environmental Sciences, Faculty of Agriculture and Forestry, University of Helsinki,
    00014 Helsinki, Finland
3   Faculty of Veterinary Medicine, Universidade Lusófona de Humanidades e Tecnologias, Campo Grande, 376,
    1749-024 Lisbon, Portugal; agusmaolima@gmail.com
*   Correspondence: isabelsousa@isa.ulisboa.pt

**Abstract:** Cereal products are staple foods highly appreciated and consumed worldwide. Nonetheless, due to the presence of gluten proteins, and other co-existing compounds such as amylase-trypsin inhibitors and fermentable short-chain carbohydrates in those products, their preference by consumers has substantially decreased. Gluten affects the small gut of people with celiac disease, triggering a gut inflammation condition via auto-immune response, causing a cascade of health disorders. Amylase-trypsin inhibitors and fermentable short-chain carbohydrate compounds that co-exists with gluten in the cereal-based foods matrix have been associated with several gastrointestinal symptoms in non-celiac gluten sensitivity. Since the symptoms are somewhat overlapped, the relation between celiac disease and irritable bowel syndrome has recently received marked interest by researchers. Sourdough fermentation is one of the oldest ways of bread leavening, by lactic acid bacteria and yeasts population, converting cereal flour into attractive, tastier, and more digestible end-products. Lactic acid bacteria acidification in situ is a key factor to activate several cereal enzymes as well as the synthesis of microbial active metabolites, to positively influence the nutritional/functional and health-promoting benefits of the derived products. This review aims to explore and highlight the potential of sourdough fermentation in the Food Science and Technology field.

**Keywords:** sourdough fermentation; lactic acid bacteria; acidification; nutritional advantages; functional properties

## 1. Introduction

Wheat and gluten-containing products have been associated with a wide range of gastrointestinal disorders, reducing their consumption worldwide and leading to considerable soaring demand for gluten-free products [1]. Indeed, wheat and other gluten-containing foods have been recognized for triggering a wide range of health problems, from which gluten intolerance observed in celiac disease (CD) is the most important [1]. CD is characterized by the small gut inflammation condition via an autoimmune response, triggered by specific gliadin peptide fractions of the gluten network proteins, which affects around 1 to 3% of the population [2] inducing mucosal inflammation, small gut villous atrophy and malabsorption of macro and micronutrients [3]. The only medical treatment available is a strict and life-long restriction of gluten-containing foods, including not only wheat but also rye and barley [4].

Nonetheless, recent evidence suggests that gluten is not the only culprit in triggering gastrointestinal disorders [1]. Many other components co-exist with gluten in wheat and gluten-containing foods, members of a short-chain carbohydrates group, named FODMAPs, an acronym that stands for Fermentable Oligo-Di-Monosaccharides And Polyols [5], that

have also have been associated with several gastrointestinal symptoms in non-celiac gluten sensitivity (NCGS), commonly known as wheat sensitive (WS) individuals, even though they do not exhibit clinical markers of CD or wheat protein sensitivity [1]. Irritable bowel syndrome (IBS) is the most common gastrointestinal disorder in NCGS individuals, affecting of around 10–15% of the population, characterized by flatulence, bloating, abdominal pain/discomfort and altered bowel habitat, that can profoundly affect the life quality of these patients [6].

The rapid fermentation of FODMAPs in the large intestine is suggested as a mechanism that triggers IBS symptoms. Amylase-trypsin inhibitors (ATI's) and other non-gluten proteins have been associated as pro-inflammatory effect compounds capable of triggering gastrointestinal symptoms in humans. However, it is still not clear whether the wheat-related symptoms are due to wheat proteins (ATI´s and gluten) or FODMAPs or even a synergic combination of both [7,8].

One method to degrade gluten proteins fractions and to decrease the amount of FODMAPs and possibly the bioactivity of ATIs in bread, are the prolonged fermentation processes in breadmaking. Sourdough fermentation is a long-term fermentation and represents one of the oldest biotechnology processes, dating back to ancient Egypt, characterized by a synergic activity between lactic acid bacteria (LAB) and yeast populations [9]. During the last century, sourdough fermentation was widely replaced by industrial fast-tracked processes, using large quantities of chemical and/or baker's yeast leavening agents [10]. Under these leavening agents, the main polymeric cereal components (e.g., proteins, starch) and short-chain carbohydrates compounds (e.g., fructans) undergo very mildly or absent hydrolysis/degradation resulting in less easily digestible foods with possible consequences to human health and life quality [11]. Compared to the other leavening agents, the sourdough can positively influence the bread sensory quality, generating more natural bread with a clean label and increasing the nutritional and functional properties [12,13].

The impact of sourdough fermentation has been associated with organic acids synthesis, the activation of the flour endogenous enzymes and the microbial secondary metabolic activity [14,15]. Along with the advantages related to the sourdough process, the increase of the in vitro protein digestibility, nutritional indexes, and amount of soluble fibre [16,17], the decrease of the glycemic index [15,18], phytate content [16,19] trypsin inhibitors, and other anti-nutritional factors reduction [8,20–22], and increments on soluble phenolic compounds correlated with antioxidant capacity enhancement [23–25] have been described (For review: Montemurro et al. [26]).

In cereal sourdough fermentation, oligopeptides are released mainly by the activity of cereal endoproteases, during primary proteolysis, whereas the release of small-sized peptides and free amino acids occurs through microbial peptidase secondary metabolic activity, especially that of LAB (lactic acid bacteria) [12,14,27]. LAB possess different enzymatic activities that can be an interesting tool to increase the free amino acids profile and generate several bioactive peptides with antimicrobial and antioxidant properties and can also modulate inflammatory processes [28–30]. This effect is strain-specific and thereby very dependent on the microorganisms used for sourdough fermentation [31–33].

In this review, the role of sourdough LAB fermentation to improve the digestibility of bakery goods, based on their impact on cereals prolamins degradation and anti-nutritional factors by synergic proteolytic activity between LAB and endogenous cereal proteases, will be described. Evidence in nutritional, functional, and health-promoting properties by LAB proteolytic activity, will also be reported.

## 2. Cereal Prolamins: Celiac Disease and Wheat Sensitivity

Wheat, barley, and rye are closely related cereals belonging to the *Triticale genus* [34]. Cereal prolamin of wheat (gliadins), barley (hordeins) and rye (secalins) are the frequent causes of food allergies and autoimmune disorders known as gluten sensitivity or intolerance. Wheat proteins induce classical inflammation conditions via immune responses that affect the skin, gut, or respiratory tract and could also induce anaphylaxis or asthma [11].

The ingestion of prolamin-containing foods is the causal factor in CD or WS individuals, leading to the atrophy of the small intestine villi [35]. The 33-mer peptide from $\alpha$-gliadin has frequently been described as the most important CD-immunogenic sequence within gluten [36].

Wheat proteins, glutenins, and gliadins, are the major storage proteins of the wheat grain: gliadins are alcohol-soluble proteins and glutenins are soluble in dilute acids [37]. The gliadins are monomeric as they contain only intramolecular disulphide bonds, and are grouped into $\alpha$-, $\gamma$- and $\gamma$-type gliadins, based on their amino acid composition. Glutenins are highly polymeric proteins, divided into high molecular weight (HMW) and low molecular weight (LMW) fractions [38]. The term prolamins refer to the proline (Pro) and glutamine (Gln) rich alcohol-soluble proteins, typically found in cereals. Prolamins are further divided into three subgroups, based on their molecular weights and sulphur contents: the HMW prolamins, s-rich prolamins, and s-poor prolamins (Table 1). Within the wheat gluten proteins, the HMW prolamins include the HMW glutenins, whereas the LMW glutenins and $\alpha$- and $\gamma$-gliadins belong to the s-rich prolamin subgroup. The s-poor prolamins include the $\omega$-gliadins. Cysteine residues are only present in $\alpha$-gliadins and $\gamma$-gliadins monomers [14,39].

**Table 1.** The prolamins of the cereal's grains (wheat, rye, barley) (Adapted from Loponen, [40]).

| | Prolamins of the Cereals Grains | | |
| --- | --- | --- | --- |
| | **Wheat** | **Rye** | **Barley** |
| HMW prolamins | HMW glutenins | HMW secalins | D-hordeins |
| S-rich prolamins | LMW glutenins | - | B-hordeins |
| S-rich prolamins | $\alpha$- and $\gamma$-gliadins | $\gamma$-secalins | $\gamma$-hordeins |
| S-poor prolamins | $\gamma$-gliadins | $\gamma$-secalins | C-hordeins |
| | Gluten proteins | Secalins | Hordeins |

## 3. FODMAPs: Non-Celiac Gluten Sensitivity and Irritable Bowel Symptoms

Short-chain dietary carbohydrates are additional components that co-exist with gluten in wheat and gluten-containing foods, considered as undigested compounds in the human small intestine but colon-fermented by microbiota bacteria, to short-chain fatty acids and gases. These components known as FODMAPs can promote beneficial effects, as dietary fibres or prebiotic effects [41], but also adverse impacts on human health [42], mainly for individuals with functional gastrointestinal disorders, as irritable bowel symptoms (IBS) [43].

Accordingly, IBS is a gastrointestinal disorder characterized by both abdominal pain and abnormal bowel habit, in which the ingestion of the FODMAPs leads to a cascade of symptoms related to bloating, distension, excessive gas production, and urgency to defecate, implying severe effects on patient's life quality [44].

FODMAPs are natural compounds present not only in wheat and gluten-containing foods but also in many other food groups and raw materials (some examples are given in Table 2). They often comprise the dietary non-digestible, osmotically active, and readily fermentable carbohydrates of galactooligosaccharides ($\alpha$-GOS), fructans and fructooligosaccharides (FOS), fructose in excess of glucose, lactose, and polyols (sugar-alcohols) (for review: Ispiryan et al. [43]).

Alfa-galactooligosaccharides ($\alpha$-GOS), are a large group of oligosaccharides and the most common polymeric FODMAP compound found in foods, being ubiquitous in pulse seeds and legumes as raffinose, stachyose and verbascose. Gastrointestinal discomfort in IBS patients as well as in healthy individuals is due to the absence of $\alpha$-galactosidase [43].

Similarity, non-digestible fructans, made up of fructose units with a single D-glucosyl unit at the end, generally is referred to as fruto-oligosaccaharides (2-9 fructose units) or oligofructose (>10 fructose units), both present mainly in grain and cereals foods [20]. Wheat and rye, especially whole grains, are the major sources of the dietary intake of

fructans [20]. Since humans lack the enzymes hydrolyzing fructans to fructose (exo- and endo- inulinase and invertase) these polymers cannot be digested and absorbed in the intestine [45]; (for review: Nyyssola et al. [46]).

The disaccharide lactose consisting of galactose and glucose molecules linked by a β (1–4) glycosidic bond, is the main FODMAPs in dairy products. A high fraction of human adults are lactose intolerant, due to the decreased intestinal lactase activity, leading to gastrointestinal disorders in many individuals [47]. Lactose can also be found in cereal-based products depending on their formulation ingredients [48]. Enzymes belonging to β-galactosidases catalyze the hydrolysis of lactose to its monosaccharide components [46].

Another group of FODMAPs that can trigger gastrointestinal symptoms are the sugar polyols (known as sugar-alcohols) generally present in stone fruits, some vegetables (e.g., sorbitol) and can also be produced during the fermentation of cereal-based products (e.g., mannitol) [49]. Mannitol is found in fruits, such as watermelon and peach as a minor component [1], and in sourdough fermentation products (as bread), due to the conversion of the fructose to mannitol through mannitol dehydrogenases by heterofermentative lactobacilli fermentation [50].

Finally, fructose is a ubiquitous monosaccharide found in a wide variety of fruits and vegetables either in free form or as a part of sucrose, linked with glucose. Fructose is regarded as FODMAPs, when it is present in excess of glucose since the intake of glucose together with fructose considerably boost their absorption [46].

All these components belong to the FODMAPs group, characterized as gastrointestinal disorders agents since they are slowly digested, or not digested at all, in the small intestine, due to certain limitations of the human digestion system, passing undigested to the colon, causing gastrointestinal discomfort in IBS patients and probably in many others called non-celiac gluten sensitivity [1].

Omitting these products from the diet should be the simplest solution to avoid gastrointestinal symptoms. However, evidence has shown that although the levels of FODMAPs in food products need to be reduced to tolerated levels, they should not be eliminated, since they act as a dietary fibre and prebiotic on the human body (especially fructans and α-GOS in particular), with beneficial effects to the gut microbiota, vital for good immunological responses and producers of several metabolites, like essential fatty acids [1,46].

**Table 2.** Examples of FODMAP containing foods [g/100 g DM].

| FODMAPs Contents [g/100 g DM] | | | | | | | |
|---|---|---|---|---|---|---|---|
| Products | Fructans | GOS | Fructose (FEG) | Lactose | Polyols Sorbitol | Polyols Mannitol | Reference |
| 1. Gluten-containing cereal | | | | | | | |
| Whole wheat | 1.88 | 0.14 | - | na | 0.04 | 0.01 | |
| Whole barley | 1.38 | 0.56 | - | na | nd | nd | |
| Rye | 3.61 | 0.13 | - | na | 0.01 | nd | Ispiryan et al. [43] |
| Spelt | 0.85 | 0.13 | - | na | nd | nd | |
| 2. Gluten-free cereals and pseudocereals | | | | | | | |
| Corn starch | nd | nd | - | na | nd | nd | |
| Potato starch | nd | nd | - | na | nd | nd | |
| Quinoa | nd | 0.09 | - | na | 0.28 | nd | Ispiryan et a. [43] |
| Buckwheat | nd | 0.01 | - | na | 0.17 | nd | |
| 3. Seeds from pulses | | | | | | | |
| Lentil | 3.98 | 1.44 | - | na | 0.95 | nd | |
| Chickpea | nd | 2.11 | - | na | nd | nd | |
| Soy | nd | 3.55 | - | na | 0.06 | nd | Ispiryan et al. [43] |
| Faba bean | nd | 3.45 | - | na | 0.03 | nd | |

**Table 2.** *Cont.*

| Products | FODMAPs Contents [g/100 g DM] | | | | | | |
|---|---|---|---|---|---|---|---|
| | Fructans | GOS | Fructose (FEG) | Lactose | Polyols Sorbitol | Polyols Mannitol | Reference |
| 4. Fruits | | | | | | | |
| Pear | nd | nd | 2.3–5.0 | na | 2.3–60 | nd | |
| Apple | nd | nd | 0.14–0.76 | na | 0.70–0.83 | nd | |
| Peach | nd | nd | 0.0–4.2 | na | 0.68–0.99 | nd | Muir et al. [1] |
| Blackberries | nd | nd | nd | na | 4.6 | nd | |
| 5. Dairy products | | | | | | | |
| Yoghurt | na | na | na | 2.9–4.2 | na | na | |
| Curd cheese | na | na | na | 1.8 | na | na | Gille et al. [51] |
| Bovine milk | na | na | na | 4.1–5.0 | na | na | |
| 6. Cereal products and gluten-free alternatives [g/100 g FW] | | | | | | | |
| White wheat bread | 0.44 | 0.01 | 0.19 | nd | 0.01 * | | |
| Wheat sourdough bread | 0.11 | nd | nd | nd | 0.21 * | | Ispiryan et al. [43] |
| Gluten-free white bread | nd | nd | nd | nd | 0.03 * | | |

FODMAPs determination via High-Performance Anion-Exchange Chromatography with Pulsed Amperometric Detection (HPAEC-PAD); Food Groups 1–5, results referred to dry matter (DM); Food group 6- results referred to fresh weight (FW); FEG–fructose in excess of glucose; nd—Not detected or values below 0.005 g/100 g DM; na—Not analyzed; * sum of polyols: xylitol; sorbitol and mannitol.

## 4. Effect of Sourdough Fermentation in Alleviating Symptoms of Celiac Disease and Wheat Sensitivity

### 4.1. Proteolytic Enzymes from Dormant and Germinated Wheat Grains

Generally, proteolytic enzymes (proteases) are grouped into proteinases and peptidases [14]. The proteases are divided into exoproteases and endoproteases: exoproteases hydrolyze only peptide bonds near the terminal ends of polypeptides, or hydrolyze small peptides, whereas endoproteases are those that cleave peptide bonds located in the central part of proteins. According to their chemical structures and active sites, the four main classes of proteinases are aspartic, cysteine, serine, and metalloproteinases [40]. Peptidases (e.g., serine carboxypeptidase), hydrolyze specific peptide bonds or completely break down peptides to amino acids [14].

Dormant wheat grains contain proteolytic activities that are derived mainly from aspartic proteinases and serine carboxypeptidases [40,52], which are activated by moisture, within a medium temperature range, under acidic and mild pH conditions, pH 3.0–3.5 and pH 4.0–4.5, respectively [53]. Both proteases are present in the endosperm and are partially associated with gluten proteins. Their activation cause changes in the gluten structures, mainly on glutenins subunits [40].

The hydrolysis of prolamins requires specific enzyme activities, and cereal grains naturally contain those specific proteases to hydrolyze gluten proteins, synthesized during the cereal grain germination. Grain's germination process induces the production of endogenous cereal enzymes, and, in general, the cysteine proteinases have been considered the most important group of proteases to hydrolyze the prolamins (especially gliadins) in germinated wheat grains. Nevertheless, the presence of aspartic protease intensifies the overall proteolysis. The cysteine and aspartic proteinase activities are strongly dependent at a pH range between 3.8–5.0 (from pH 3.8 the total proteolytic activities are higher) [53].

Previous studies have shown that cysteine proteinases in wheat grain are capable to hydrolyze both gliadins and glutenins [54], whereas the wheat aspartic proteinases predominantly degrade just glutenins [53,55].

In addition to the cysteine proteinase, serine and metalloproteinase activities also occur in germinated wheat grains [40]. Both these proteases operate efficiently at pH values close to neutral (6.0–8.0). Of all the peptidases, carboxypeptidases are active under

mildly acidic conditions and probably can be suitable to hydrolyze proline-containing toxic peptides prolamins as well as derived peptides [14], although low specific activities were achieved [56].

*4.2. Prolamin Proteolysis in Wheat Sourdough Fermentation*

In the second half of the last century, fast leavening processes by chemicals and/or baker's yeast almost replaced the use of sourdough. Using these leavening agents, the main polymeric cereal components (e.g., proteins) undergo very mildly or no hydrolysis during processing.

Sourdough fermentations offer nearly ideal conditions for the degradation of cereal prolamins since it is a pH-dynamic semi-fluid system that allows the activation and stimulation of cereal proteases, according to its optimal-pH range. Additionally, microbial acidification during fermentation increases the solubility of the prolamins, which makes them more susceptible to proteolytic breakdown [15].

Evidence from sourdough applications has shown the efficiency of this system to degrade the gluten network since this proteolytic system efficiently degrades glutenins as well as the gliadin fraction [57].

The degradation of the toxic 33-mer peptide (considered the most immunogenic peptide responsible for triggering celiac disease) [58] have been reported to be possible by strain-specific and thereby very dependent on the sourdough LAB strains used [32].

Additionally, surveys based on extensive prolamin hydrolysis showed that when wheat germinated grains were used as raw materials in sourdough fermentation, considerable proteolysis occurred, in which, around 95% of the prolamins were hydrolyzed [14,53]. Previous studies have shown that cysteines proteinases of germinated wheat grain, activated by sourdough fermentation, were capable to hydrolyze both gliadins and glutenins [53–55,59], whereas the wheat aspartic proteinases predominantly degrade only glutenins [40].

The hydrolysis of glutenins, during sourdough fermentation, results in depolymerization and subsequent solubilization [14], being mainly dependent on the acidification by pH changes (14–15). LMW glutenins are partially hydrolyzed during sourdough fermentation and the degradation of HMW glutenins is virtually quantitative [40,57,60].

Loponen et al. [53] reported that when germinated wheat grains, with their high and diverse proteolytic activities, were used as a raw material in sourdough fermentation, extensive prolamins proteolysis occurred after six hours of fermentation with LAB, resulting in a virtual disappearance of the protein bands in the alcohol-soluble fraction.

These studies evidence the concept that a combination of germinated grain and sourdough fermentation can probably be used to hydrolyze prolamins to levels that might be better tolerated by wheat sensitivity patients.

The quantification of the prolamin contents in sourdoughs samples using the R5-ELISA method confirmed the observations from the SDS-PAGE analyses), where the prolamin concentration of the germinated wheat sourdough fermentation decreased significantly, during the first 6 h of fermentation (27,000 ppm to 3700 ppm). After 12 h of germinated wheat sourdough dough fermentation, the prolamin content was only 1200 ppm, whereas in the control sourdough the prolamin concentration remained at 24,000 ppm [53].

However, it is worth noting, according to the upper limit of gluten content in food, that it would not be safe for patients with CD, as the daily gluten intake should be between 10–20 parts per million of gluten [1]. Even though the sourdough application can contribute to an extensive gluten network degradation, the levels reached are not enough to ensure them as safe products for celiac individuals' diets.

In this work, evidence that the toxicity of wheat prolamins, or their hydrolysis products, was reduced or eliminated by using germinated raw material in sourdoughs, was not investigated. Nevertheless, Hartmann et al. [61] showed that the pool of proteases presents in germinated grains, including wheat, hydrolyzed typical gliadins peptides into fragments that were not harmful to celiac patients. Mandile et al. [62], also performed a

clinical study in young people with CD utilizing sourdough products produced from the combination of selected LAB to degrade wheat gluten, in which no immune responses or clinical symptoms from celiac patients over 60 days, was registered.

These studies suggested that sourdough by selected LAB can be used as an important pre-digested prolamins system, making the IgE-binding proteins more degradable by digestion enzymes [12,62–64], improving the digestibility of the gluten proteins.

### 4.3. Combining Cereal Endogenous Enzymes and LAB Sourdough Fermentation

The synergic proteolytic activity between cereal endogenous proteases (primary proteolysis) and strain-specific intracellular peptidases from LAB, which provides several proline-specific peptidases (secondary proteolysis), have been associated with the complete degradation of gluten during sourdough fermentation [27,60,63,64].

Several researchers [14,15,60,65] have investigated the sequence of proteolysis events and elucidated the contributions of cereal and bacteria peptidases, that occur during the sourdough system (Figure 1).

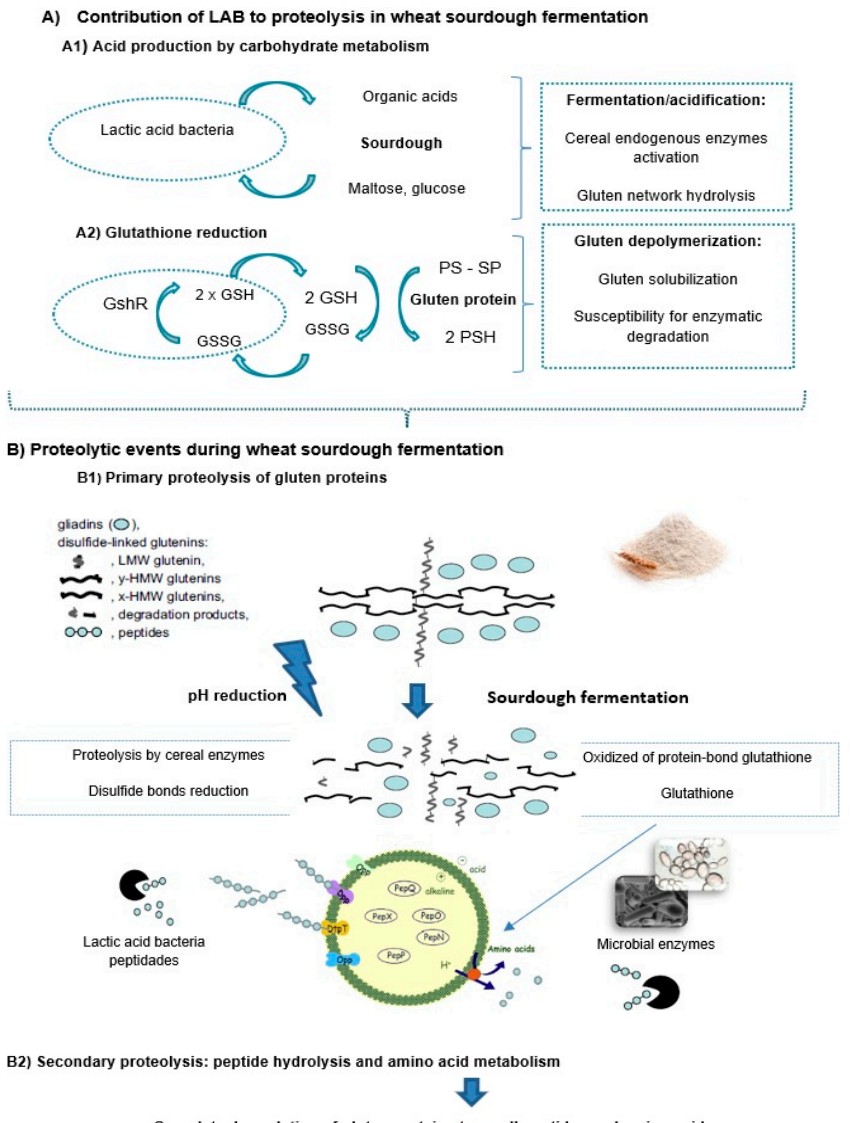

**Figure 1.** Representation of LAB contribution to gluten protein proteolysis during sourdough fermentation: (**A**) Contribution of LAB in initial stages to environment acidification (A1), glutathione reductase and related thiol-active enzymes (A2); (**B**) Proteolytic events during sourdough fermentation

based on primary proteolysis generated by the acidification and the reduction of disulfide linkages of gluten proteins by sourdough LAB, promoting the primary activity of cereal endogenous proteases (B1) and secondary proteolysis by intracellular peptidases of sourdough LAB, which complete the proteolysis of gluten proteins, liberating free amino acids (B2) (adapted from Ganzle et al. [14,15]; Gobbetti et al. [12]. The representation of gluten macro polymers is based on Wieser [57]. Abbreviations: GrhR, glutathione reductase, GSH and GSSG, reduced and oxidized glutathione, respectively, PSH and PS-SP, oxidized and reduce inter-or-intramolecular disulphide bridges in gluten proteins, respectively.

Taking all factors into consideration, this proteolytic system is probably mediated by the following key enzymes and metabolic activities: the acidification by sourdough LAB shifts the dough pH to around 6.0 to 3.5–4.0, matching the ideal environmental conditions to the main enzymes in primary proteolysis, the cereal aspartic (dormant grains) and cysteine proteinases (in case of germinated grains) [12,14,52]. In addition, the acidification promoted by LAB contributes to reducing the disulfide linkages of glutenins, leading to gluten depolymerization, consequently increasing their solubility, making them more susceptible to enzymatic degradation [28].

Furthermore, the glutenins macropolymer is strongly affected by reducing agents, where the glutathione is the most important, since it undergoes thiol-exchange reactions with gluten proteins, decreasing the intermolecular disulfide cross-linking [66]. LAB also express glutathione reductase during growth in sourdough, reducing extracellular oxidized glutathione (GSSG) to reduced glutathione (GSH) [15,67,68].

This primary cereal endogenous proteolytic activity of gluten proteins generates different sized polypeptides [12,14]. Intracellular peptidases of sourdough LAB will probably complete the proteolysis of these end-products by a secondary proteolytic activity: through a complex system of ABC and ATP transporters, namely Opp (oligopeptide permease), DtpT (Di- and tripeptide permease) and Dpp (Dipeptide permease), the polypeptides will, across the cytoplasmatic membrane of LAB, liberate amino acids and microbial metabolites to the extracellular environment [12,69].

According to De Angelis et al. [69], the secondary proteolysis is mediated by the combined activity of five peptidases (PepN-aminopeptidase type N, PepO- endopeptidase, PEP-prolyl endopeptidyl peptidase, PepX- X-prolyl dipeptidyl aminopeptidase and PepQ—Prolinase) in which, all together were responsible for the degradation of the 33-mer peptide (the most important CD-immunogenic peptide from $\alpha$-gliadin sequence within gluten) within 14 h of incubation.

The concentration of free amino acids will increase, which, in turn, will suffer additional catabolic reactions by the same microorganisms [12–15,65,70]. Therefore, the sourdough LAB system has been considered a potential bioprocess to improve the nutritional and bioactive properties of bakery goods. The synthesis of health-promoting metabolites through the native cereal proteolysis represents an interesting tool to the biological fortification of the bread, by essential amino acids, bioactive and antioxidant peptides [14,70].

*4.4. Contribution of Sourdough Fermentation to Nutritional, Functional, and Human Health-Promoting Benefits*

Sourdough-like fermentation, carried out by LAB within its microbial consortium, has been largely reported as a powerful tool to enhance nutritional and functional properties in flours [12,16,71] and wheat-based foods like the bread [17,72].

The combined effect of the cereal grain germination and microbial fermentation have been well-recognized for the biological fortification of bakery goods, including gluten-free bread, by increasing the content of essential free amino acids (FAA) and considerable improvements in protein digestibility and nutritional indexes [71,73].

LAB use polypeptides to meet their demand for complex nitrogen [14], and the analysis of peptide and amino acid levels in wheat sourdoughs indicated that they uptake these nitrogen sources to grow in the sourdough system [25].

Recent studies [25,26,74,75] have shown a considerable increase in total FAA and improvements in the in vitro protein digestibility, via protein proteolysis and polypeptides solubilization by sourdough fermentation. Among FAA, a considerable increase in y-aminobutyric acid (GABA), a non-protein amino acid that is primarily produced from the decarboxylation of L-glutamic acid, was found. GABA possesses well-known physiological functions such as neurotransmission, induction of hypotension, diuretic, and tranquillizer effects [76–78].

These LAB proteolytic systems have been recognized as a potential tool not only to improve the nutritional and functional benefits of the baking goods [75,79] but also with considerable positive effects on health [12,15]. Specific strains of LAB, possessing different enzymatic activities, are recognized as a suitable approach to generate several bioactive peptides associated with different biological roles on human health [80], which generally increase during food fermentation by LAB [81].

Recent developments focused on the accumulation of (bioactive) peptides and amino acid metabolites in dough and bread from sourdough LAB fermentation, with antioxidant, anti-inflammatory properties, and cancer-preventing activities, being associated with LAB peptidase activities [82]. The ability of sourdough LAB activities to generate bioactive peptides through proteolysis of native cereal proteins has been well demonstrated [28,83,84].

Rizzello et al. [85] showed a remarkable increase in the concentration of the anticancer peptide lunasin, a fragment of the larger 2S-albumins, the most widely studied peptide for its anticancer activities, by the fermentation of whole wheat flours with sourdough LAB. Therefore, LAB is recognized as the most useful microorganisms for bioactive peptides production in fermented foods [86,87]. Regarding these studies, the interest for the selection of LAB strains, as starter cultures, to the manufacture of healthier leavened baked goods is increasing [74,88,89].

Sourdough fermentation is also considered an effective tool for starch degradation and carbohydrates metabolism [15]. The presence of low pH values of around 3.5–4.0, promoted by LAB acidification activity [12,87] and the combination of protein-rich foods like yoghurt [90,91] and fibre sources [92] demonstrated to be a potential system to reduce the glycemic index of the baking goods.

Fois and coworkers [93] showed that the application of the sourdough system can decrease substantially the glycemic index to lower values (GI < 55) while improving the quality and shelf-life of fresh pasta. Similar results were reported by Wang et al., [24] on the influence of in situ dextran produced by *Weissella confusa*, during sourdough fermentation, on technological and nutritional properties of whole-grain pearl millet bread.

Additionally, considerable reduction (about 80%) of glycemic index on experimental sourdough bread, compared to baker's yeast bread, was achieved by Nionelli et al. [94]. This effect was attributed to biological acidification by LAB, which is one of the main factors that decrease starch hydrolysis rate and index [71]. Lactic acid, which is an organic acid synthesized by LAB activity, was identified as one of the main causes to reduce starch digestibility in the human body. Lactic acid seems to affect starch digestion by lowering the $\alpha$-amylase enzymatic activity [53]. Furthermore, the chemical changes promoted upon sourdough fermentation may impact negatively on starch gelatinization performance it can increase the levels of resistant starch, which is not enzymatically digestible, therefore, with no impact on the glycemic index [95,96].

LAB enzymatic activities can contribute to increasing the soluble fibres and solubilization of the insoluble fibre fraction, correlated with the delay of starch digestion and absorption rates impacting on glycemic and insulinemic responses decrease [26–97]. Moreover, the generated peptides from native protein proteolysis, amino acids as well as free phenolic compounds, which are liberated during sourdough fermentation, seem to have a crucial role in glucose metabolism regulation, consequently, lowering the GI [18,95].

Furthermore, sourdough fermentation can decrease the content of bound phenolic compounds increasing their bioavailability, which has been correlated with in vitro antioxidant capacity improvement [95]. Recently, Wang et al. [24] and Jiang et al. [25] reported that

fermentation enhanced the solubilization of bound phenolic compounds with a consequent increase of the soluble compounds' fraction, possibly correlated with the higher DPPH radical scavenging activity achieved. These findings were consistent with those reported by Shumoy et al. [98], who emphasized that such increments might be a result of microbial acidification, and consequent activation of endogenous cereal enzymes and production of hydrolytic enzymes of LAB, during fermentation. Since the free phenolic compounds are more bioavailable the health benefits are potentially boosted [24]. In line with these findings, Bei et al. [99] found that the improvement of phenolic composition and their bioactivity can not only contribute to the antioxidant capacity but also inhibit the α-amylase and α-glucosidases performance, affecting its activity on starch hydrolysis. Additionally, digestive enzyme inhibitors from some polyphenol compounds are found to be promising approaches to help maintain a low GI diet, especially for starch-rich foods [24,100,101].

It is well-known that phytic acid is an abundant anti-nutritional factor, mainly located in the bran fraction of the whole grain's flours, strongly reducing the mineral bioavailability, due to the chelating complexes formed with mineral and trace elements. Nevertheless, it has been demonstrated that the fermentation process with LAB can efficiently degrade the phytate complex thanks to the activation of endogenous and microbial phytases [102], and successfully overcame its detrimental effect on the mineral availability [16,19,26].

Other bioactive compounds, such as anticancer, anti-inflammatory, and immunomodulatory peptides have largely been found on innovative sourdough-based products [72,79,103].

## 5. The Role of Sourdough to Reduce FODMAPs Compounds

The sourdough system has also been exploited to produce low-FODMAPs products since it was demonstrated a high potential to lower the quantity of the most indigestible oligosaccharides (mainly fructans and α-GOS) to levels that can be tolerated by NCGS and IBS individuals [1]. Some of the research works done in FODMAPs reduction by sourdough fermentation are given in Table 3.

**Table 3.** Degradation of fructans and α-GOS from foods by sourdough fermentation.

| Product/Subtract | Method Applied | FODMAP Reduction | Reference |
|---|---|---|---|
| Whole wheat bread | Fermentation of 4.5 h, 30 °C using bakery´s yeast (*Saccharomyces cerevisiae*) | 90% of fructans and raffinose | Ziegler et al. [50] |
| Whole rye bread | Sourdough fermentation rye bread (not specified) Traditional bakery´s yeast rye bread | 62% in fructans 32% in fructans | Andersson et al. [104] |
| Wheat bread | Bakey´s yeast fermentation of 180 min, 35 °C | 40% in fructans | Gélinas et al. [105] |
| Whole wheat bread | Bakery´s yeats and *K. marxianus* fermentation of 180 min, 30 °C | 95% in fructans | Struyf et al. [106] |
| Seed Beans flour (*Phaseolus vulgaris*) | Natural fermentation | 100% Raffinose | Granito et al. [107] |
| Black Beans flour (*Phaseolus vulgaris*) | Fermentation by *Lactobacillus casei* and *Lactobacillus plantarum* | 88.6% raffinose | Granito and Álvarez [108] |
| Soy milk (*Glycine max*) | Fermentation by *Lactobacillus rhamnosus* 6013 | 100% raffinose | Liu et al. [109] |
| Soy milk (*Glycine max*) | Fermentation by Kefir starter culture (Clerici Sacco) | 100% raffinose | Bau et al. [110] |

| Product/Subtract | Method Applied | FODMAP Reduction | Reference |
|---|---|---|---|
| Soy milk (*Glycine max*) | Fermentation by *Lactobacillus acidophilus, Bifidobacterium animalis* and *Streptococcus thermophilus* | 40% raffinose | Battistine et al. [111] |
| Faba bean flour (*Vicia faba*) | Fermentation by *Weissella cibaria*, Weissella confusa, Pediococcus pentosaceus Leuconostoc kimchi | 100% raffinose, 84% verbascose | Rizzello [112] |
| Chickpea flour (*Cicer arietinum*), Sprouted Lentil flour (*Lens culinaris*) | *Fermentation by Lactobacillus rossiae, Lactobacillus plantarum and Lactobacillus sanfrancensis* | 95% raffinose | Montemurro et al. [65] |

LAB produces lactic and acetic acids lowering the dough pH, which allows the activation of a few specific enzymes, suitable to reduce the FODMAP compounds [46,113].

In addition, other important baking conditions cannot be excluded, as the proofing time, temperature, and the types of microorganism used in sourdough, to achieve a major impact to lower the FODMAPs levels in the final products [1,113].

However, the greater factor to reduce these compounds content below the "cutoff" line in sourdough bread is the proofing time [46,48].

Ziegler and coworkers [48] have demonstrated the importance of the long-fermentation time, showing that a proofing time of 4.5 h was suitable to reduce FODMAP content (fructans and raffinose) of around 90% in whole wheat bread and 77% in spelt bread. Additionally, the authors emphasized that a shorter proofing time of around 1 h lead to an increase in fructose since it is a result of the breakdown of fructans.

Like gluten sourdough degradation, the selection of specific bacteria strains is the key to producing a final bread with low-FODMAP compounds, and with no negative impact on optimal quality attributes (e.g., bread volume). Andersson and coworkers [105] have shown lower fructans content (62% decrease) in rye bread prepared by sourdough method than in traditional baker´s yeast-leavened rye bread (only 32% decrease).

This is most likely due to the action of bacterial hydrolytic enzymes, but it is also possible that endogenic enzymes are activated at the lower values of pH achieved in sourdough systems. Gelinas et al. [105] reported that around 40% of wheat fructans can be degraded during baking by *S. cerevisiae.* However, Struyf et al. [106] claimed that these levels of fructans reduction are not enough for individuals with functional gastrointestinal disorders by FODMAP compounds, as IBS patients. Struyf and coworkers [107] demonstrated that the combination of an inulinase-secreting yeast, *Kluyveromyces marxianus*, with *Saccharomyces cerevisiae* can reduce significantly, of around 90%, the fructans in dough made from whole wheat flour, whereas only 56% of fructans were degraded by *S. cerevisiae*.

A recent survey performed by Loponen and coworkers [114], showed that the strain belonging to *Lactobacillus crispatus*, was also capable to hydrolyze fructans. From this work, a potential problem emerged, since the fructose release by the fructans hydrolysis during sourdough can be converted to mannitol by heterofermentative LAB, which is also a FODMAP compound. However, Loponen and Ganzle, [50] suggested that the mannitol can also be reduced by the activity of specific *lactobacilli*.

As described above, the α-GOS are ubiquitous in plant seeds and a major source of anti-nutritional compounds mainly found in legumes [115]. Based on the increased role of legumes and pulses in modern diets, namely as a source of protein, there is a crucial interest to reduce this oligosaccharide in food.

Considering the abundance expression of the α- galactosidase in bacteria and fungi, sourdough fermentation can be an attractive alternative to degrade this antinutritional factor [107], and in part generate novel flavours [116] and textures [117].

Soybean is worldwide considered the commercial plant containing higher amounts of α-GOS. Nonetheless, the almost complete reduction of raffinose and stachyose from soy milk by specific LAB fermentation have been well reported in previous studies [109–111].

LAB fermentation has also been applied to entire seeds and pulses, and promising results were reported by Granito and Alvarez [108], in which, significant removal of around 90% in raffinose in whole soybeans by LAB fermentation was obtained. Several studies have reported a considerable degradation of α-GOS on faba bean fermentation, using different LAB [107,112]. Furthermore, Montemurro et al. [26], showed a significant reduction of raffinose (95%) in chickpeas and lentils by fermentation using different LAB.

Indeed, evidence has shown that sourdough baking reduces and converts FODMAPs in the rye and wheat flour, but the extent of FODMAP reduction is dependent on the nature of the fermentation organisms, the conditions of the fermentation process, as well as the raw material [50].

Despite the lack of support from clinical trials, sourdough-derived products are likely to play a significant role when developing healthier baking products for individuals with non-celiac clinical symptoms as patients who suffer from IBS [8].

## 6. Conclusions and Future Perspectives

This review highlighted the interesting scientific research work that has been done by several researchers, exploring the potential of sourdough LAB fermentation in the Food Science and Technology field, demonstrating its positive impact on Food Nutrition and Human Health.

As an overview, sourdough fermentation is considered the most traditional and effective tool to improve the nutritional and functional value of baking products, answering modern consumers´ demand for health-promoting products, representing a new opportunity for the food industry [26,70,118].

Taking all into consideration, future efforts should focus on the implementation and optimization of the sourdough LAB process at the industrial scale, as a promising bioprocess alternative to match consumer demand, responding well to the needs of modern consumers and healthy-market niches.

Additionally, considering the new era of food processing based on more natural and sustainable production and food waste reduction, sourdough fermentation can be an alternative approach to be exploited using whole grain flours, with great potential and impact on circular economy and the ecological footprint.

Therefore, future efforts should be focused on targeting the optimization of sourdough bioprocesses selecting specific lactic acid bacteria and yeasts, depending on the functional/nutritional characteristics of the raw material and those desired in the food product.

**Author Contributions:** C.G. and I.S. conceived and planned the experiments; C.G. performed all samples preparation and analysis, data analysis, interpretation of the results assisted by I.S., A.R. and A.L. and C.G. wrote the manuscript; A.L. revised the manuscript; A.R. and I.S. supervised and validated the work, data discussion and writing—review and editing. All authors have read and agreed to the published version of the manuscript.

**Funding:** The publication was funded by the FCT through the research unit UID/AGR/04129/2020 (LEAF).

**Institutional Review Board Statement:** Not applicable.

**Informed Consent Statement:** Not applicable.

**Data Availability Statement:** The data presented in this study is available upon reasonable request.

**Acknowledgments:** The first author thanks the Portuguese Foundation for Science and Technology (FCT) to have funded the research work through the research Unit UID/AGR/04129/2020 (LEAF).

**Conflicts of Interest:** The authors declare no conflict of interest.

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
