# Peer review of "Sourdough Fermentation as a Tool to Improve the Nutritional and Health-Promoting Properties of Its Derived-Products"

_fermentation, doi:10.3390/fermentation7040246_

Round 1

Reviewer 1 Report

General comments

This review is valuable and interesting and deals a topic worthy of further study.

The authors in the manuscript “Sourdough fermentation as a tool to improve the nutritional and health-promoting properties of its derived-products” provide a balanced and in depth view of the topic.

They reviewed the bibliography related to the subject and made detailed the importance of sourdough fermentation by lactic acid bacteria to improve the digestibility of bakery goods. Evidence in nutritional, functional, and health-promoting properties by LAB proteolytic activity, were also be reported.

English language should be carefully revised, some spelling and grammar mistakes have been found throughout the manuscript.

In my opinion, the review is worth to be published after some little modifications. Here a few suggestions.

Line 66-67: To support the statement I suggest you to add the reference Reale et al., 2013 (doi: 10.1111/1750-3841.12206).

Line 369-371: I suggest you to add the appropriate and recent references on the importance of the selection process of LAB strains with promising potential for the production of different types of sourdough-based products. Lactic acid bacteria, in fact, play an important role during sourdough fermentation for the development of flavor components other than nutritional and health-promoting properties of the finished baked products.  Reale et al., 2020 (https://doi.org/10.1016/j.lwt.2020.110092) and Di Renzo et al., 2018 (doi: 10.3389/fmicb.2018.00429).

Line 377-378: I suggest to add more recent reference. I suggest Demirkesen-Bicak et al., 2021 (https://doi.org/10.3390/foods1003051).

Line 392- 393: I suggest to add a reference for this statement. You could add this reference Messia et al., 2016 (https://doi.org/10.1016/j.jcs.2016.03.004) regarding the positive influence of LAB fermentation exerted on the solubilization of fiber fraction.

Author Response

Dear Reviewer,

We are thankful for your contribution to improving our review manuscript.

All your suggestions were very welcome and considered as important improvements to give additional support to the statements. The changes made are yellow highlighted throughout the manuscript.

Sincerely,

Carla Graça

Reviewer 2 Report

This review is well written, clearly described and useful to readers .

The paper “Sourdough fermentation as a tool to improve the nutritional and health-promoting properties of its derived-products” deals with an interesting topic on the role of sourdough LAB fermentation to improve the digestibility of bakery goods, based on their impact to cereals prolamins degradation and anti-nutritional factors by synergic proteolytic activity between LAB and endogenous cereal proteases.  This work, in addition to being interesting, is of great relevance and also applicability as it reports potential to solve the problem of the consumption of cereals, given the great diffusion of people intolerant to gluten. The article is well organized and explained in detail, clearly described in all the sections and easy to follow from the reader. This review is highly exhaustive and topical and it reports current information on the topic, highlighting interesting scientific research work done by several researchers on the potential of sourdough LAB fermentation in food science and technology field, offering also interesting research insight. Conclusions are consistent with the review argument, reporting also the future perspectives, focusing on targeting the optimization of sourdough bioprocesses selecting specific lactic acid bacteria and yeasts. The paper merits to be published on Fermentation.

The title needs to be corrected in the word nutritional instead of nutritional

Author Response

Dear Reviewer,

We are much obliged for your revision contribution and the pleasant comments on our review work.

The title was corrected in the word "Nutritonal"  to "Nutritional", as pointed out.

Sincerely,

Carla Graça